

# A model of twenty-three metabolic-related genes predicting overall survival for lung adenocarcinoma

Zhenyu Zhao,  Boxue He,  Qidong Cai,  Pengfei Zhang,  Xiong Peng,
Yuqian Zhang,  Hui Xie and  Xiang Wang

Department of Thoracic Surgery, The Second Xiangya Hospital of Central South University, Central South
University, Changsha, Hunan, China
Hunan Key Laboratory of Early Diagnosis and Precise Treatment of Lung Cancer, The Second Xiangya
Hospital of Central South University, Central South University, Changsha, Hunan, China

## ABSTRACT

**Background**. The highest rate of cancer-related deaths worldwide is from lung adenocarcinoma (LUAD) annually. Metabolism was associated with tumorigenesis and cancer development. Metabolic-related genes may be important biomarkers and metabolic therapeutic targets for LUAD.

**Materials and Methods**. In this study, the gleaned cohort included LUAD RNA-SEQ data from the Cancer Genome Atlas (TCGA) and corresponding clinical data ($n = 445$). The training cohort was utilized to model construction, and data from the Gene Expression Omnibus (GEO, GSE30219 cohort, $n = 83$; GEO, GSE72094, $n = 393$) were regarded as a testing cohort and utilized for validation. First, we used a lasso-penalized Cox regression analysis to build a new metabolic-related signature for predicting the prognosis of LUAD patients. Next, we verified the metabolic gene model by survival analysis, C-index, receiver operating characteristic (ROC) analysis. Univariate and multivariate Cox regression analyses were utilized to verify the gene signature as an independent prognostic factor. Finally, we constructed a nomogram and performed gene set enrichment analysis to facilitate subsequent clinical applications and molecular mechanism analysis.

**Result**. Patients with higher risk scores showed significantly associated with poorer survival. We also verified the signature can work as an independent prognostic factor for LUAD survival. The nomogram showed better clinical application performance for LUAD patient prognostic prediction. Finally, KEGG and GO pathways enrichment analyses suggested several especially enriched pathways, which may be helpful for us investigative the underlying mechanisms.

Corresponding author
Xiang Wang, wangxiang@csu.edu.cn

## INTRODUCTION

Lung cancer (LC) is one of the most common cancers worldwide and the main cause of cancer-related mortality (*Bray et al., 2018*; *Torre et al., 2015*). Non-small cell lung cancer (NSCLC) accounts for 85% of all LCs. The 5-year survival rate after the diagnosis of LC is 15.6% (*Nanavaty, Alvarez & Alberts, 2014*). In NSCLC, lung adenocarcinoma (LUAD) is

the major histological subtype (*Balzer et al., 2018*), and the recurrence rate and mortality rate remain high despite recent advances in surgical methods, neoadjuvant therapies, and immunotherapies.

As bioinformatics advances in oncology research, researchers can utilize access public resources from multiple public databases such as The Cancer Genome Atlas (TCGA) and the Gene Expression Omnibus (GEO), as well as Surveillance and Epidemiology and End Results (SEER) (*Doll, Rademaker & Sosa, 2018*; *Li et al., 2018*; *Liu et al., 2019*). Bioinformatics has contributed to determining the prognosis and treatment of LC (*Parikh, 2019*). There have been numerous studies on gene prognosis models that could contribute to the selection of LC treatment methods and the prediction of survival after LC surgery; for example, a prognostic signature containing six genes (RRAGB, RSPH9, RPS6KL1, RXFP1, RRM2, and RTL) to evaluate the prognosis of NSCLC patients (*Xie & Xie, 2019*). In another article on prognostic characteristics of LUAD, a prognostic model based on 20 genes was developed to predict patient overall survival (OS) (*Zhao, Li & Tian, 2018*). These prognostic signatures all have better clinical application performance.

Metabolic changes in LC are the key to diagnosis, and metabolic remodelling is a critical factor in tumorigenesis and development (*Chen et al., 2019b*). Metabolic remodelling not only provides substances and energy for the survival and proliferation of tumour cells but also protects tumour cells so that they can survive, proliferate, and metastasize in harsh microenvironments (*Hensley et al., 2016*). Therefore, changes in metabolism affect tumour prognosis and treatment effects (*Chang, Fang & Gu, 2020*; *Cruz-Bermúdez et al., 2019*). To explore the correlation between metabolic genes and the prognosis of LUAD patients, we utilized the TCGA-LUAD database to build a prognostic signature of multiple metabolic-related genes and validated it in GEO data sets for LUAD patients. We conducted this study, and our findings suggested that metabolic-related gene signatures may be a prognostic marker for LUAD patients.

## MATERIALS AND METHODS

### Data collections

First, we got clinical information for patients with LUAD from TCGA (https://portal.gdc.cancer.gov/). It included 497 LUAD patients with mRNA expression profiles and clinical follow up information was available for our study. The number of obtainable clinical cases for the selected subjects was 445 after removing 52 patient samples from the study due to a lack of clinical information (such as survival time, T stage, N stage, and so on) or survival time less than 30 days (avoiding non-cancer-related death samples). A total of 445 LUAD patients and their information were utilized to build a training cohort for identifying prognostic metabolic-related genes and building a prognostic risk model. Next, we downloaded the LUAD gene expression data from GEO (https://www.ncbi.nlm.nih.gov/geo/) in two accessed datasets GSE30219 and GSE72094. Removing other cancer pathological types, such as lung squamous cell carcinoma, 83 LUAD samples, and 393 LUAD samples were utilized to build a testing cohort for validating the prognostic value of the TCGA-LUAD prognostic risk model (up to April 01, 2020).

## Identification of metabolic-related genes in TCGA-LUAD

First, we obtained 944 hub metabolism-related genes from the intersection of the MSigDB database (https://www.gsea-msigdb.org/gsea/msigdb) and TCGA-LUAD. Then, a Wilcoxon signed-rank test was performed on normal and cancer tissues in the training cohort by "limma" R package ($|\log FC| > 0.5$; FC: fold change; a false discovery rate (FDR) $P < 0.05$) (*Diboun et al., 2006*). The heatmap was plotted by the "pheatmap" R package and we obtained 336 metabolic-related differentially expressed genes. Second, after univariate Cox regression analysis, 59 metabolic genes were retained ($P < 0.05$) by using the method that the correlation between expression values of metabolic genes and survival of samples in the training cohort. Last, we performed lasso-penalized Cox regression analysis to identify more important metabolic genes for OS prediction through the "glmnet, survival" R package (*Zhang et al., 2019*). We obtained twenty-three metabolic-related genes for risk model building. The three-step screening method was robust and performed via Perl (https://www.perl.org/) and R (version 3.6.1).

## Building the prognostic metabolic gene signature

To construct the prognostic model, we utilized lasso-penalized Cox regression analysis to select the prognostic metabolic-related gene (*Tibshirani, 1997*). We obtained a risk score for each patient by their coefficient. Risk score= (Coef AKR1A 1× expression of AKR1A1) + (Coef NT5E × expression of NT5E) + ……(Coef TYMS × expression of TYMS) (*Liu et al., 2020*). R software packages "survival" and "survminer" were used to calculate the optimal cut-off value for risk scores and plot Kaplan–Meier survival curves (*Chan et al., 2018*). Using the median as a point of differentiation, we differentiated patients into two groups: high-risk and low-risk. The R package "survivalROC" was used to plot time-dependent ROC curves for predicting the diagnostic value (*Heagerty, Lumley & Pepe, 2000*). The concordance index (C-index) was used to evaluate the predictive ability of the risk model.

## Verification of the prognostic signature as an independent risk factor and correlation analysis between the clinical characteristics and risk scores

Patients with complete information on the corresponding clinical data were available for univariate and multivariate analysis. $P < 0.05$ symbolizes statistically significant (*Liu et al., 2020*). We performing the Student's $t$-test to verify the correlation between clinical characteristics and risk scores.

## Construction and verification of the predictive nomogram

The nomogram was built by the "rms" R package according to training cohort data (*Iasonos et al., 2008*). In our study, the tumour-node metastasis (TNM) model and the prognostic signature were integrated into the predictive nomogram. We performed the calibration plot and C-index to investigate the predictive ability of the nomogram. The calibration plot was used to assess whether the numerical value of the predicted value of the model and the probability of the occurrence of the ending event were consistent (*Fenlon et al., 2018*). We used C-index to assess the predictive ability of the nomogram. It estimates the probability

that the predicted result is consistent with the actual observed result. We compared the TNM model, prognostic model, and the nomogram model through ROC analysis and C-index (*Liu et al., 2020*). Next, we verified the nomogram by C-index, ROC analysis, and calibration plot in the testing cohort data.

### KEGG and GO pathways enrichment analyses

To study the biological role of mRNA markers in LUAD patients, we utilized the Kyoto Encyclopedia of Genes and Genomics (KEGG) and Gene Ontology (GO) pathway enrichment analysis to explore which pathways the differentially expressed genes were mainly enriched in (up to April 11, 2020). Gene Set Enrichment Analysis (GESA) (https://www.gsea-msigdb.org/gsea/index.jsp) was utilized to find enriched terms in the training cohort or testing cohort (*Subramanian et al., 2005*). We choose "c2. cp. kegg. v6.2. symbols. gmt. gene sets" as a reference gene set from the MSigDB database (https://www.gsea-msigdb.org/gsea/msigdb/). $P < 0.05$, FDR $q$-value $< 0.25$, and normalized enrichment score $|NES| \geq 1$ suggested statistically significant. We plotted the results by "ggplot2, gridExtra, grid, plyr" R package. All operations are carried out in GSEA_4.0.3. GO pathway enrichment analysis of metabolic genes was performed by "clusterProfiler, org.Hs.eg.db, plot, ggplot2" in R $p$ ackage (*Pathan et al., 2015*).

### Statistical analyses

logFC: logarithmic value of FC; positive/negative logFC indicates the logarithmic foldness of upregulation/downregulation; $|\log FC| > 0.5$ indicates multiple differences in the gene expression greater than 0.5 between normal tissues and cancer tissues. A coefficient is a number that expresses a measurement of a particular quality of a substance or object under specified conditions (*Bøvelstad et al., 2007*). The risk score was calculated according to the formula:

$$\text{Risk score}(\text{patients}) = \sum_n \Big(\text{coefficient}(\text{mRNAn}) * \text{expression}(\text{mRNAn})\Big).$$

C-index had a lower accuracy from 0.50–0.70, medium accuracy between 0.71–0.90, and higher accuracy when greater than 0.90 degrees (*Kim, Schaubel & Mccullough, 2018*). The Area Under Curve (AUC) value of the time-dependent ROC lay in the range of 0.5–0.9 viewed as statistically significant. All statistical analyses were conducted through R 3.6.1.

## RESULTS

### Construction of the prognostic signature from the training cohort

The workflow of the study was shown in Fig. 1. In our study, 445 LUAD patients (Table S1, clinical information in Table S1) from the TCGA data set were assigned to the training sample cohort; 83 LUAD patients (Table S3, clinical information in Table S4) and 393 LUAD patients (Table S5, clinical information in Table S6) from the GEO data set were assigned to the testing sample cohort for batch processing. After the Wilcoxon signed-rank test was applied to the training set, we obtained 336 meaningful metabolic genes (Fig. 2, Table S7). From the 336 meaningful metabolic genes, we got 59 mRNAs which were considered to be significantly associated with OS in LUAD patients, 42 high-risk genes,

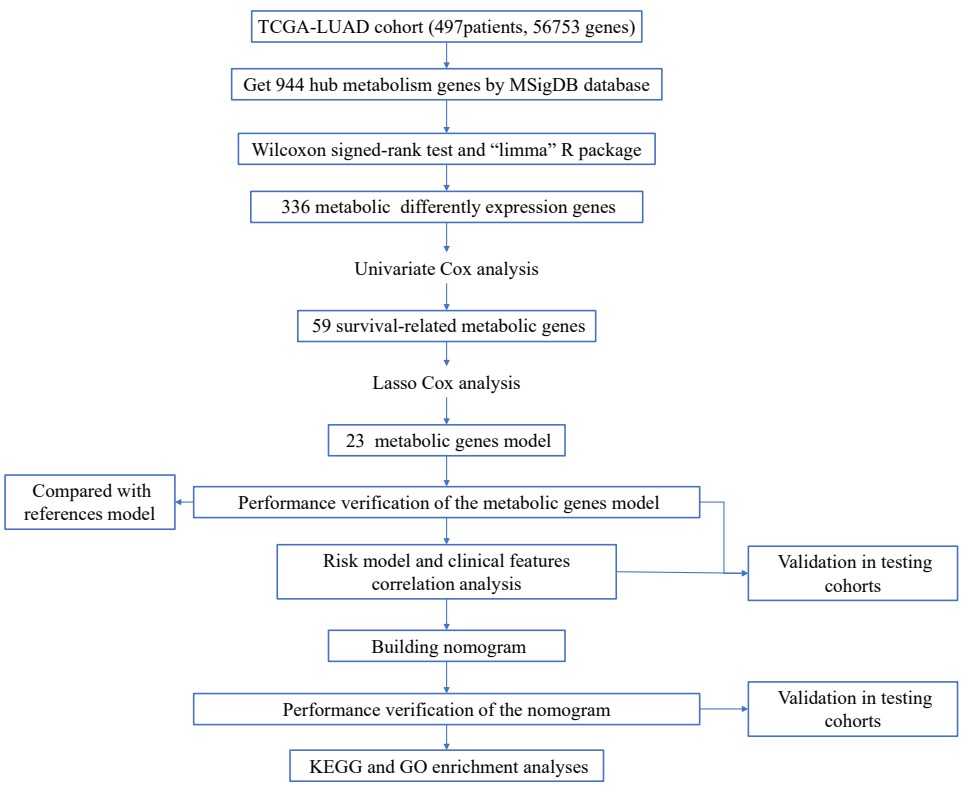

**Figure 1  Workflow of the study.**

and 17 low-risk genes (Fig. 3, Table S8). Finally, lasso-penalized Cox analysis identified 23 genes (AKR1A1, NT5E, PTGIS, GMPS, MBOAT1, ADCY9, B4GALT1, MAOB, INPP4B, NEU1, ALDOA, ENTPD2, GNPNAT1, GSTA3, PKM, HK3, ALDH2, AK2, LDHA, CHPT1, SMS, CTPS2, and TYMS) to construct the prognostic model (Table 1).

## The twenty-three metabolic genes signature and predictability assessment in the training cohort

Finding that there is a significant and independent correlation between the expression of twenty-three prognostic mRNAs and OS, we believed that combining the 23 metabolic genes to form a twenty-three gene signature could predict a patient's prognosis. According to the optimal cut-off of 1.53, we classified training cohort samples into two groups: a high-risk group ($n = 222$) and a low-risk group ($n = 223$) (Table S9). The OS rate between the two risk groups was significantly different ($P = 5.543e-10$) (Fig. 4A).

The AUC was 0.798, 0.747, and 0.734 for the 1, 3, and 5-year OS, separately (Fig. 4B). We plotted patients' risk curves in the training cohort and analysed their distribution in Figs. 4D–4E, and the heat map reveals the prognostic mRNA expression patterns between two distinct prognostic patient groups (Fig. 4C).

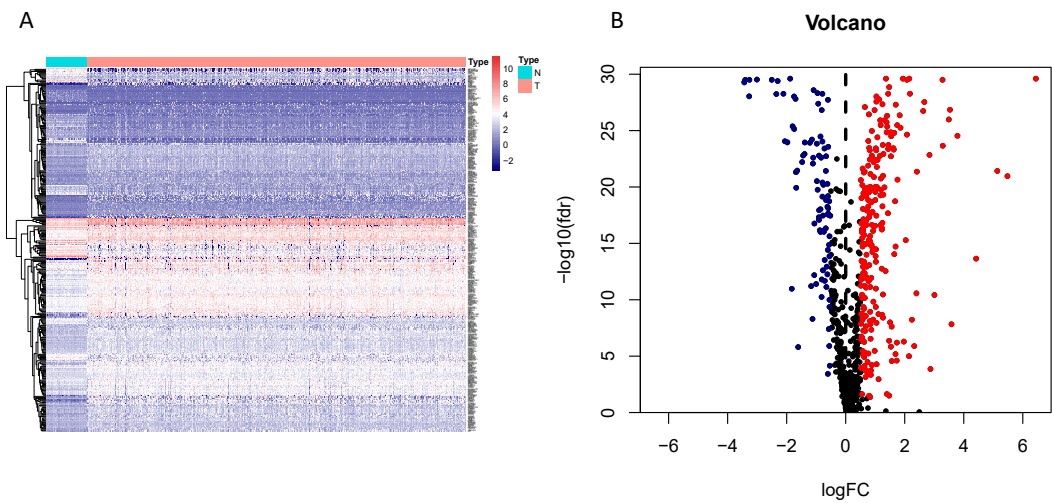

**Figure 2** **Heatmap and volcano map of the differentially expressed genes in normal and tumour tissues from TCGA-LUAD.** (A) Heatmap of the differentially expressed genes in normal and tumour tissues from TCGA-LUAD; (B) Volcano map of the differentially expressed genes in normal and tumour tissues from TCGA-LUAD. In the volcano map, red: genes upregulated in tumour groups; yellow: genes downregulated in the tumour group; black: no differentially expressed genes in the tumour group.

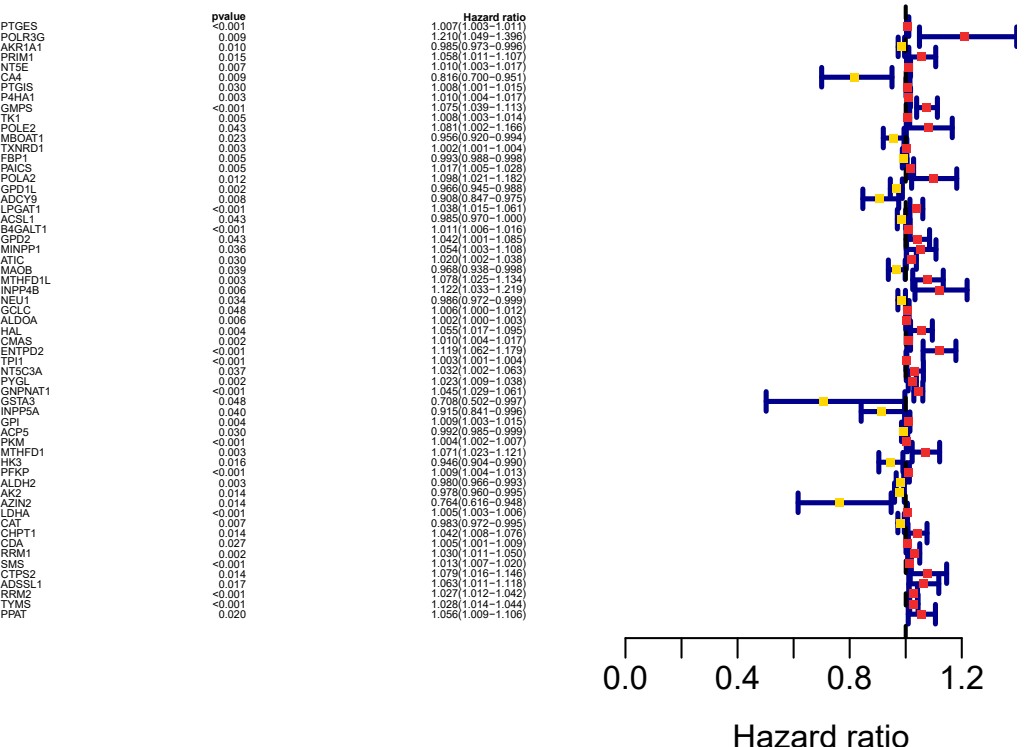

**Figure 3** **Forest map of the fifty-nine mRNAs (red: high-risk genes; yellow: low-risk genes).**

**Table 1** The 23-gene signatures screened by their coefficients.

| Gene | Coefficient | HR | HR.95L | HR.95H | P |
|---|---|---|---|---|---|
| AKR1A1 | −0.005 | 0.985 | 0.973 | 0.996 | 0.01 |
| NT5E | 0.003 | 1.01 | 1.003 | 1.017 | 0.007 |
| PTGIS | 0.01 | 1.007 | 1.003 | 1.011 | 0.001 |
| GMPS | 0.003 | 1.075 | 1.039 | 1.113 | 4.17E−05 |
| MBOAT1 | −0.025 | 0.956 | 0.92 | 0.994 | 0.023 |
| ADCY9 | −0.009 | 0.908 | 0.847 | 0.975 | 0.008 |
| B4GALT1 | 0.002 | 1.011 | 1.006 | 1.016 | 3.05E−06 |
| MAOB | −0.019 | 0.968 | 0.938 | 0.998 | 0.039 |
| INPP4B | 0.046 | 1.122 | 1.033 | 1.219 | 0.006 |
| NEU1 | −0.003 | 0.986 | 0.972 | 0.999 | 0.034 |
| ALDOA | 0.001 | 1.002 | 1 | 1.003 | 0.006 |
| ENTPD2 | 0.065 | 1.119 | 1.062 | 1.179 | 2.25E−05 |
| GNPNAT1 | 0.028 | 1.045 | 1.029 | 1.061 | 2.86E−08 |
| GSTA3 | −0.084 | 0.708 | 0.502 | 0.997 | 0.048 |
| PKM | 0.001 | 1.004 | 1.002 | 1.007 | 0 |
| HK3 | −0.008 | 0.946 | 0.904 | 0.99 | 0.016 |
| ALDH2 | −0.001 | 0.98 | 0.966 | 0.993 | 0.003 |
| AK2 | −0.013 | 0.978 | 0.96 | 0.995 | 0.014 |
| LDHA | 0.002 | 1.005 | 1.003 | 1.006 | 4.20E−09 |
| CHPT1 | 0.024 | 1.042 | 1.008 | 1.076 | 0.014 |
| SMS | 0.001 | 1.013 | 1.007 | 1.02 | 0 |
| CTPS2 | 0.052 | 1.079 | 1.016 | 1.146 | 0.014 |
| TYMS | 0.016 | 1.028 | 1.014 | 1.044 | 0.0002 |

**Notes.**
HR, hazard ratio.

## Validation of the twenty-three metabolic gene signatures

To test the robustness of the prognostic signature, according to the risk score cut-off of the training cohort, GSE30219 was structured into a high-risk group ($n = 25$) and a low-risk group ($n = 58$) (Table S10); and GSE72094 was structured into a high-risk group ($n = 196$) and a low-risk group ($n = 197$) (Table S11). In the GSE30219 cohort, the Kaplan–Meier survival curves of the prognostic signature have a statistically significant difference in the two predicted risk groups (Fig. 5, $P = 1.176e−02$). The AUC was 0.694, 0.645, and 0.637 for the 1, 3, and 5-year OS, separately (Fig. 5B). The risk curves and expression of twenty-three metabolic genes in the GSE30219 cohort were shown in Figs. 5C–5E *Fig 5C-E*. And in the GSE72094 cohort, Kaplan–Meier survival curves of the prognostic signature has a statistically significant difference in the two predicted risk groups (Fig. 6, $P = 1.417e−10$). The AUC was 0.695, 0.725, and 0.742 for the 1, 3, and 5-year OS, separately (Fig. 6B). The risk curves and expression of the twenty-three metabolic genes in the GSE30219 cohort were shown in Figs. 6C–6E. We also compared our results with the two published gene signature studies (*Xie & Xie, 2019*; *Zhao, Li & Tian, 2018*) and found that the C-index results were better than their signatures (Table 2).

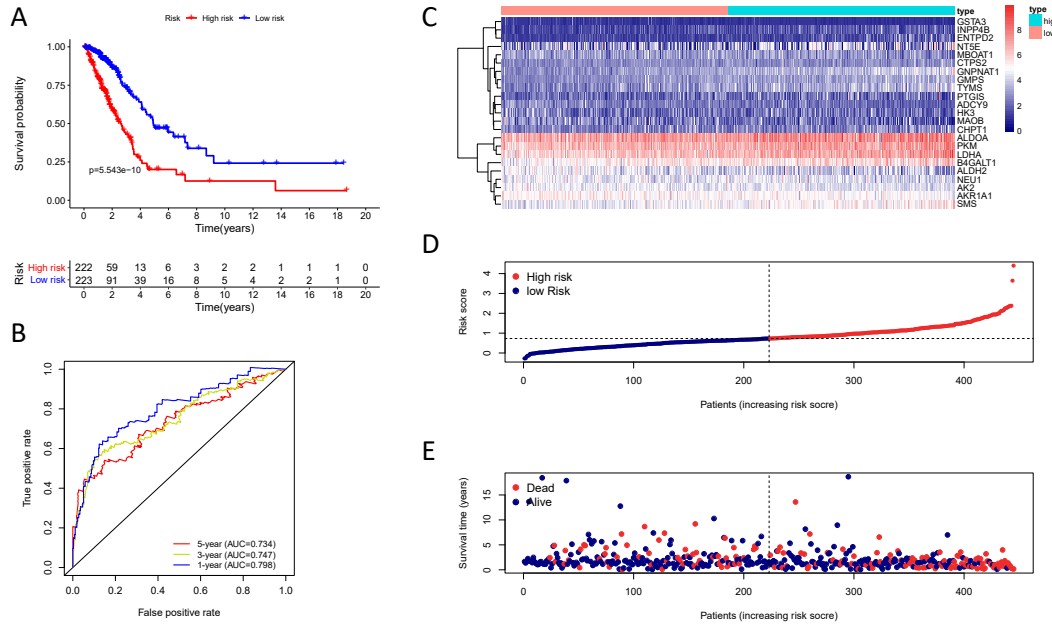

**Figure 4  Kaplan–Meier analysis, time-dependent ROC analysis, and risk score analysis for the twenty-three gene signature in the TCGA-LUAD cohort.** (A) Kaplan–Meier curve of the twenty-three gene signature in the TCGA cohort; (B) time-dependent ROC analysis of the twenty-three gene signature in the TCGA cohort; (C–E) a heatmap of mRNA expression of the twenty-three gene signature, and risk curves in the TCGA cohort.

Taken together, these results indicated a better predictive performance about our prognostic signature.

## Correlation analysis between the prognostic signature and clinical characteristics

445 patients with their information from TCGA-LUAD cohort were utilized for the correlation analysis. Being male ($P = 0.017$) and TNM stage ($P < 0.01$) have a significant correlation with a higher risk score. Samples with higher T, N, and M grading were also significantly correlated with a higher risk score (Table 3).

83 patients with their information from GSE30219 cohort were utilized for the correlation analysis. TNM stage was significantly associated with a higher risk score ($P < 0.001$). Samples with higher T, N, and M grading were also significantly correlated with a higher risk score ($P < 0.01$) (Table 3). And 393 patients with their information from GSE72094 cohort were utilized for the correlation analysis. TNM stage was significantly correlated with a higher risk score ($P < 0.001$) (Table 3).

## Validation of the independent prognostic factor

We analysed 445 patients, with a median age of 68, grouping them by gender (Table 4). The results identified that our prognostic signature was an independent OS prognostic indicator (Figs. 7A and 7B). We also verified that the prognostic signature can serve as an independent prognostic indicator in the GSE30219 cohort (Figs. 7C and 7D) and

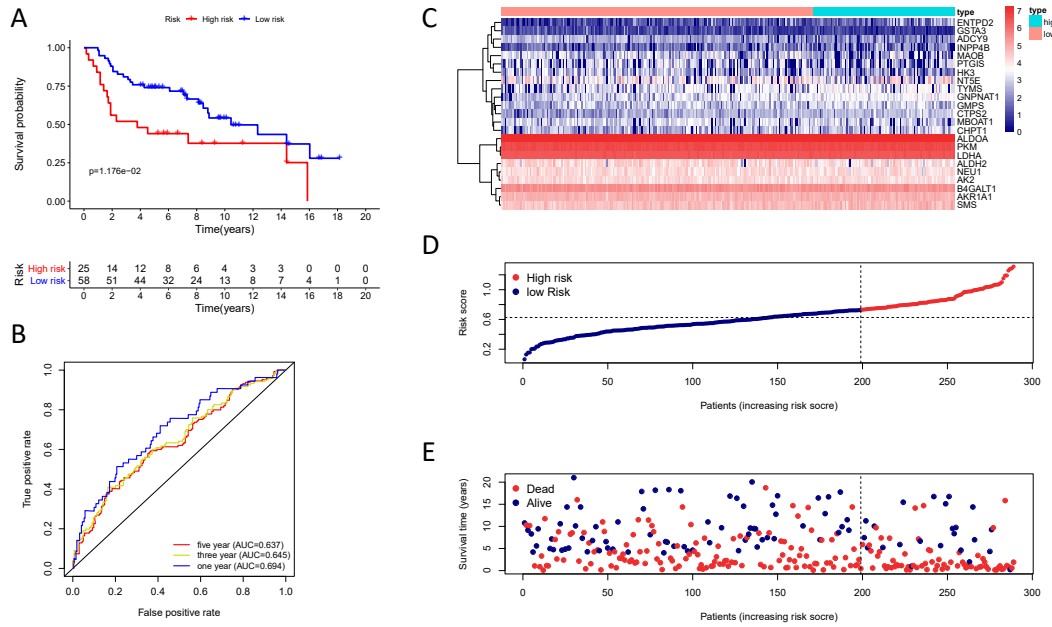

**Figure 5  Kaplan–Meier analysis, time-dependent ROC analysis, and risk score analysis for the twenty-three gene signature in the GSE30219 cohort.** (A) Kaplan–Meier curve of the twenty-three gene signature in the GSE30219 cohort; (B) time-dependent ROC analysis of the twenty-three gene signature in the GSE30219 cohort; (C–E) a heatmap of mRNA expression of the twenty-three gene signature, and risk curves in the GSE30219 cohort.

GSE72094 cohort (Figs. 7E and 7F). The multivariate Cox analysis indicating that the prognostic signature was significantly associated with OS in each cohort when adjusted for the TNM stage (Table 4). And stratification analysis showed that high-risk group was significantly correlated with a poorer OS (Figs. 8A–8F). However, in the GSE30219 testing cohort, patients in the high-risk group from TNM stages III and IV show no significantly correlated with OS (Fig. 8D).

## Construction and verification of the predictive nomogram

We built the nomogram by including the independent prognostic roles (Fig. 9A). Calibration plots verified the performance of the nomogram (Fig. 9B). The C-index of the TNM model, prognostic signature, and nomogram model were 0.654, 0.730, and 0.793, separately (Table 5). The AUC was 0.838, 0.785, and 0.779 for the 1, 3, and 5-year OS, respectively. The nomogram model showed a better AUC predicting 1, 3, and 5-year OS than the prognostic model in the training cohort (Figs. 9C–9E).

Next, we verified the clinical application of the nomogram in the GSE30219 and GSE72094 cohorts. In the GSE30219 and GSE72094 testing cohorts, calibration plots verified the performance of the nomogram (Figs. 10A and 10E). In the GSE30219 testing cohort, the C-index of the TNM model, prognostic signature, and nomogram model were 0.612, 0.682, and 0.684 (Table 5). The AUC was 0.731, 0.686, and 0.722 for the 1, 3, and 5-year OS, separately (Figs. 10B–10D). In the GSE72094 testing cohort, the C-index of the TNM model, prognostic signature, and nomogram model were 0.579, 0.709, and 0.713,
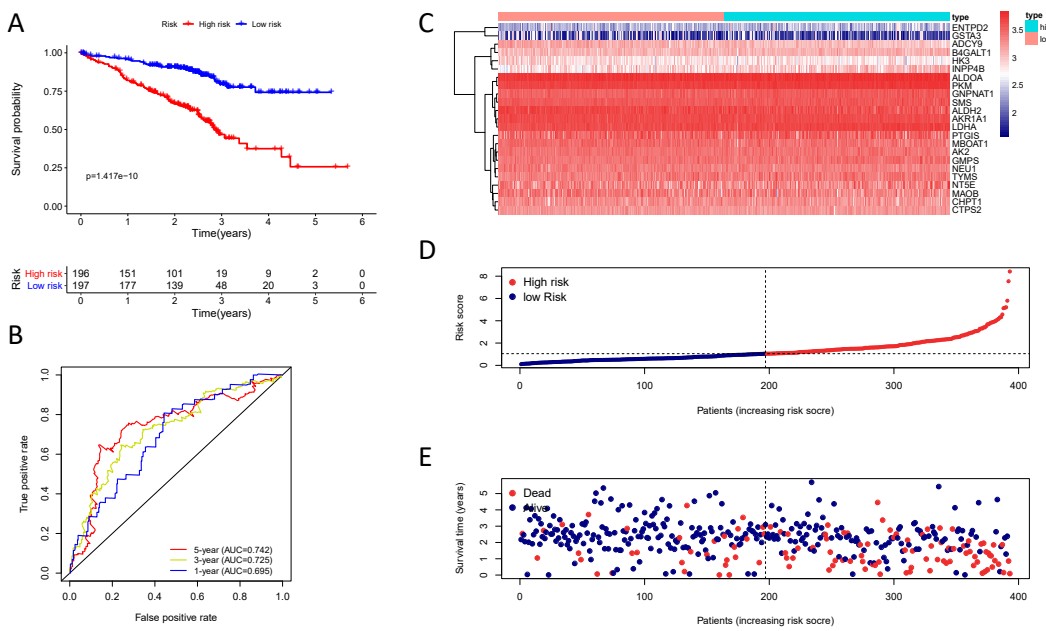

**Figure 6** **Kaplan–Meier analysis, time-dependent ROC analysis, and risk score analysis for the twenty-three gene signature in the GSE72094 cohort.** (A) Kaplan–Meier curve of the twenty-three gene signature in the GSE72094 cohort; (B) time-dependent ROC analysis of the twenty-three gene signature in the GSE72094 cohort; (C–E) a heatmap of mRNA expression of the twenty-three gene signature, and risk curves in the GSE72094 cohort.

separately (Table 5). The AUC was 0.708, 0.794, and 0.771 for the 1, 3, and 5-year OS, separately (Figs. 10F–10H). The nomogram model identified the better AUC predicting 1, 3, and 5-year OS than the prognostic signature in both testing cohorts.

Taken together, the nomogram model increased the predicting ability of the prognostic signature. These results indicated the better predictive performance of the nomogram model.

## KEGG and GO pathways enrichment analyses

KEGG enrichment analyses suggesting that a majority of the metabolism-related pathways such as the metabolism of fatty acid, arachidonic acid, glycerophospholipid, alpha-linolenic acid, and pyrimidine were associated with the low-risk group, while the cell cycle, mismatch repair, ubiquitin-mediated proteolysis, and p53 signalling pathways were associated with the high-risk group (Fig. 11A, Table 6, Table S12). Besides, we performed GO pathway enrichment analysis on 23 genes. These 23 genes were statistically significant in metabolic-related pathways, such as nucleotide biosynthetic processes, secretory granule lumen, and monosaccharide binding in Biological Processes (BP), cellular component (CC), and molecular function (MF) (Fig. 11B, Table S13).

Zhao et al. (2020), *PeerJ*, DOI 10.7717/peerj.10008

Peerj

**Table 2  Comparison of the twenty-three gene prognostic signature to the two published prognostic signatures.**

| Studies | TCGA cohort | | | GSE30219 cohort | | | GSE72094 cohort | | |
|---|---|---|---|---|---|---|---|---|---|
| | HR (95% CI) | P | C-index | HR (95% CI) | P | C-index | HR (95% CI) | P | C-index |
| Present study 23-gene signature | 4.55(3.487~5.946) | 2.00E−16 | 0.730 | 3.80(1.977~7.308) | 6.25E−05 | 0.682 | 1.74(1.514~1.99) | 2.96E−15 | 0.709 |
| Zhao, K.et al. 20-gene signature | 1.10(1.073~1.123) | 6.20E−16 | 0.697 | 1.98(1.683~2.319) | 1.72E−19 | 0.682 | 1.53(1.39~1.672) | 2.17E−15 | 0.704 |
| Xie, H.et al. 6-gene signature | 1.29(1.167~1.428) | 7.45E−07 | 0.632 | 2.34(1.825~2.996) | 1.87E−11 | 0.668 | 1.82 (1.506~2.206) | 7.10E−10 | 0.660 |

**Notes.**

HR, hazard ratio; CI, confidence interval.

Peer J

**Table 3 Correlation analysis of the clinical characteristics and the twenty-three gene signature in LUAD.**

| Characteristics | TCGA training cohort (n = 445) | | | | | GSE30219 testing cohort (n = 83) | | | | | GSE72094 testing cohort (n = 393) | | | | |
| --- | --- | --- | --- | --- | --- | --- | --- | --- | --- | --- | --- | --- | --- | --- | --- |
| | n | Mean (rick score) | SD | t | P | n | Mean (rick score) | SD | t | P | n | Mean (rick score) | SD | t | P |
| Age(years) | | | | | | | | | | | | | | | |
| <68 | 247 | 0.765 | 0.64 | −0.164 | 0.87 | 23 | 0.543 | 0.346 | −0.088 | 0.931 | 153 | 1.253 | 0.93 | −0.524 | 0.600 |
| ≥68 | 198 | 0.775 | 0.663 | | | 60 | 0.55 | 0.272 | | | 240 | 1.305 | 1.04 | | |
| Gender | | | | | | | | | | | | | | | |
| Female | 244 | 0.702 | 0.611 | −2.393 | 0.017 | 19 | 0.535 | 0.357 | −0.183 | 0.856 | 219 | 1.203 | 1.012 | −1.850 | 0.065 |
| Male | 201 | 0.851 | 0.686 | | | 64 | 0.552 | 0.274 | | | 174 | 1.389 | 0.972 | | |
| TNM stage | | | | | | | | | | | | | | | |
| Stage I+II | 348 | 0.68 | 0.584 | −4.822 | <0.001 | 60 | 0.456 | 0.235 | −6.598 | <0.001 | 320 | 1.166 | 0.842 | −3.771 | <0.001 |
| Stage III+IV | 97 | 1.083 | 0.765 | | | 23 | 0.875 | 0.239 | | | 73 | 1.805 | 1.392 | | |
| T | | | | | | | | | | | | | | | |
| T1+2 | 387 | 0.716 | 0.595 | −3.5 | 0.001 | 73 | 0.528 | 0.299 | −2.479 | 0.024 | – | – | – | – | – |
| T3+4 | 58 | 1.125 | 0.86 | | | 10 | 0.697 | 0.185 | – | – | – | | | | |
| N | | | | | | | | | | | | | | | |
| N0 | 292 | 0.71 | 0.639 | −2.629 | 0.009 | 71 | 0.502 | 0.276 | −4.03 | 0.001 | – | – | – | – | – |
| N1-3 | 153 | 0.881 | 0.657 | | | 12 | 0.817 | 0.245 | | | – | – | – | | |
| M | | | | | | | | | | | | | | | |
| M0 | 403 | 0.742 | 0.64 | −2.577 | 0.013 | 67 | 0.468 | 0.241 | −5.959 | <0.001 | – | – | – | – | – |
| M1 | 42 | 1.028 | 0.689 | | | 16 | 0.881 | 0.251 | – | – | – | | | | |

**Notes.**

SD, standard deviation; TNM, tumornode metastasis.

**Table 4** Univariate and multivariate Cox regression analysis of overall survival in each cohort.

| Variables | Univariate analysis | | | | Multivariate analysis | | | |
|---|---|---|---|---|---|---|---|---|
| | HR | HR.95L | HR.95H | *P* | HR | HR.95L | HR.95H | *P* |
| | | | TCGA training cohort (*n* = 445) | | | | | |
| Age(≥68/ <68) | 0.999 | 0.98 | 1.018 | 0.922 | 1.007 | 0.987 | 1.028 | 0.491 |
| Gender (female/male) | 1.006 | 0.692 | 1.462 | 0.976 | 0.936 | 0.638 | 1.373 | 0.736 |
| TNM stage (I+II/III+IV) | 1.659 | 1.404 | 1.961 | 3.01E−09 | 2.677 | 0.994 | 2.831 | 0.043 |
| T | 1.592 | 1.278 | 1.982 | 3.27E−05 | 1.009 | 0.788 | 1.291 | 0.944 |
| N | 1.798 | 1.463 | 2.209 | 2.34E−08 | 1.142 | 0.714 | 1.826 | 0.58 |
| M | 1.798 | 0.985 | 3.283 | 0.056 | 0.466 | 0.116 | 1.871 | 0.281 |
| Risk Score | 3.89 | 2.925 | 5.174 | 9.79E−21 | 3.699 | 2.717 | 5.036 | 9.80E−17 |
| | | | GSE30219 testing cohort (*n* = 83) | | | | | |
| Age(≥68/ <68) | 1.037 | 1.023 | 1.052 | 3.22E−07 | 1.031 | 0.994 | 1.07 | 0.102 |
| Gender(female/male) | 1.815 | 1.129 | 2.917 | 0.014 | 1.126 | 0.491 | 2.586 | 0.779 |
| TNM stage(I+II/III+IV) | 2.719 | 0.762 | 3.03 | 2.35E−03 | 2.409 | 1.014 | 2.882 | 0.044 |
| T | 1.663 | 1.448 | 1.911 | 7.10E−13 | 1.361 | 1.126 | 1.644 | 0.001 |
| N | 1.777 | 1.51 | 2.091 | 4.36E−12 | 1.358 | 1.087 | 1.698 | 0.007 |
| M | 2.856 | 1.17 | 6.97 | 0.021 | 2.357 | 0.958 | 5.797 | 0.062 |
| Risk Score | 3.726 | 1.429 | 9.714 | 0.007 | 2.26 | 1.168 | 4.374 | 0.016 |
| | | | GSE72094 testing cohort (*n* = 393) | | | | | |
| Age(≥68/<68) | 1.007 | 0.988 | 1.027 | 0.479 | 0.999 | 0.978 | 1.019 | 0.889 |
| Gender(female/male) | 1.547 | 1.065 | 2.246 | 0.022 | 1.487 | 1.011 | 2.189 | 0.044 |
| TNM stage | 1.625 | 1.360 | 1.941 | <0.001 | 1.607 | 1.333 | 1.938 | <0.001 |
| Risk Score | 1.736 | 1.514 | 1.990 | <0.001 | 1.646 | 1.431 | 1.894 | <0.001 |

**Notes.**

HR, hazard ratio; TNM, tumornode metastasis.

## DISCUSSION

Lung cancer has the highest mortality rate among cancer-related diseases worldwide (*Bray et al., 2018*). LUAD is the principal type of LC, with a percentage of more than half of morbidity and mortality in this group of patients (*Jemal et al., 2017*). With the increasing exploration of cancer metabolic heterogeneity, metabolic genes can work as a prognostic signature for LUAD. Identification of metabolism-related gene preferences and dependence mechanisms in tumour regulation has become increasingly important (*Peng et al., 2017*). Metabolic changes in LC are strategic to the diagnosis and influence the prognosis and response to treatment (*Cruz-Bermúdez et al., 2019*). TCGA and GEO databases already have a large amount of RNA-seq data from tumour samples in multiple cancers. Prognostic signatures of LUAD have been built and developed utilizing the public databases (*Shang et al., 2017*). Several metabolic genes for LUAD, such as TKT, ALDOA, TSC1, and CYP2A6, have been demonstrated to be related to the OS of LUAD (*Lin et al., 2011*; *Wassenaar et al., 2015*). Therefore, we investigated the relationship between tumour metabolism-related genes and the prognosis of LUAD. We first built twenty-three metabolic-related gene prognostic signatures, which may be helpful for the diagnosis, treatment, and prognosis of LUAD.

Zhao et al. (2020), *PeerJ*, DOI 10.7717/peerj.10008

**Table 5** **Comparison of the nomogram model with the TNM model and prognostic model.**

| Cohort | Nomogram model | | | TNM model | | | Prognostic model | | |
|---|---|---|---|---|---|---|---|---|---|
| | HR (95% CI) | *P* | C-index | HR (95% CI) | *P* | C-index | HR (95% CI) | *P* | C-index |
| TCGA cohort | 2.719(2.269~3.259) | 2.00E−16 | 0.793 | 2.957(2.015~4.339) | 3.03E−08 | 0.654 | 4.55(3.487~5.946) | 2.00E−16 | 0.730 |
| GSE30219 cohort | 2.958(2.189~3.375) | 7.55E−16 | 0.684 | 1.267(0.889~1.803) | 0.016 | 0.619 | 3.80(1.977~7.308) | 6.25E−05 | 0.682 |
| GSE72094 cohort | 2.718(2.181~3.388) | 7.55E−16 | 0.713 | 2.606 (1.736~3.914) | 3.85E−06 | 0.579 | 1.74(1.514~1.99) | 2.96E−15 | 0.709 |

**Notes.**

TNM, tumornode metastasis; CI, confidence interval.

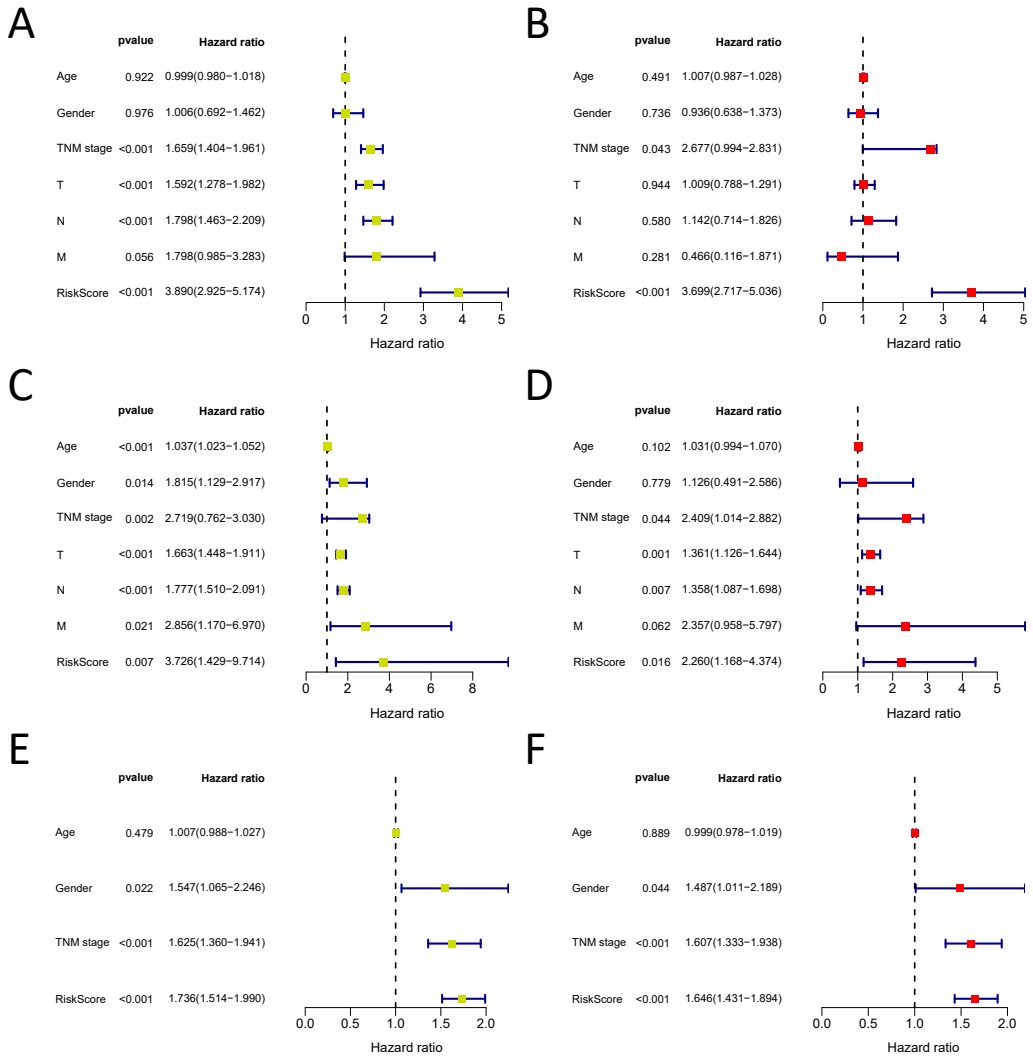

**Figure 7 Forrest plot of the univariate (yellow) and multivariate (red) Cox regression analysis in the each cohort.** (A–B) Forrest plot of the univariate (yellow) and multivariate (red) Cox regression analysis in the TCGA-LUAD cohort; (C–D) Forrest plot of the univariate (yellow) and multivariate (red) Cox regression analysis in the GSE30219 cohort; (E–F) Forrest plot of the univariate (yellow) and multivariate (red) Cox regression analysis in the GSE72094 cohort.

In this study, we identified an efficient twenty-three metabolic-related gene prognostic model based on the TCGA-LUAD databases. Our prognostic model had better performance in predicting patients' prognosis. We also verified the performance of the prognostic signature in the testing cohort, and the results confirm the prognostic value of the prognostic signature. It is worth noting that in the GSE30219 databases, the high-risk group did not show a significant correlation with poorer survival in TNM stages III and IV when compared with the low-risk group. One possible reason is that with improvements in diagnostic technology, more and more early patients are diagnosed at TNM stages I and II, and fewer patients are diagnosed at TNM stages III and IV. The AUC of the

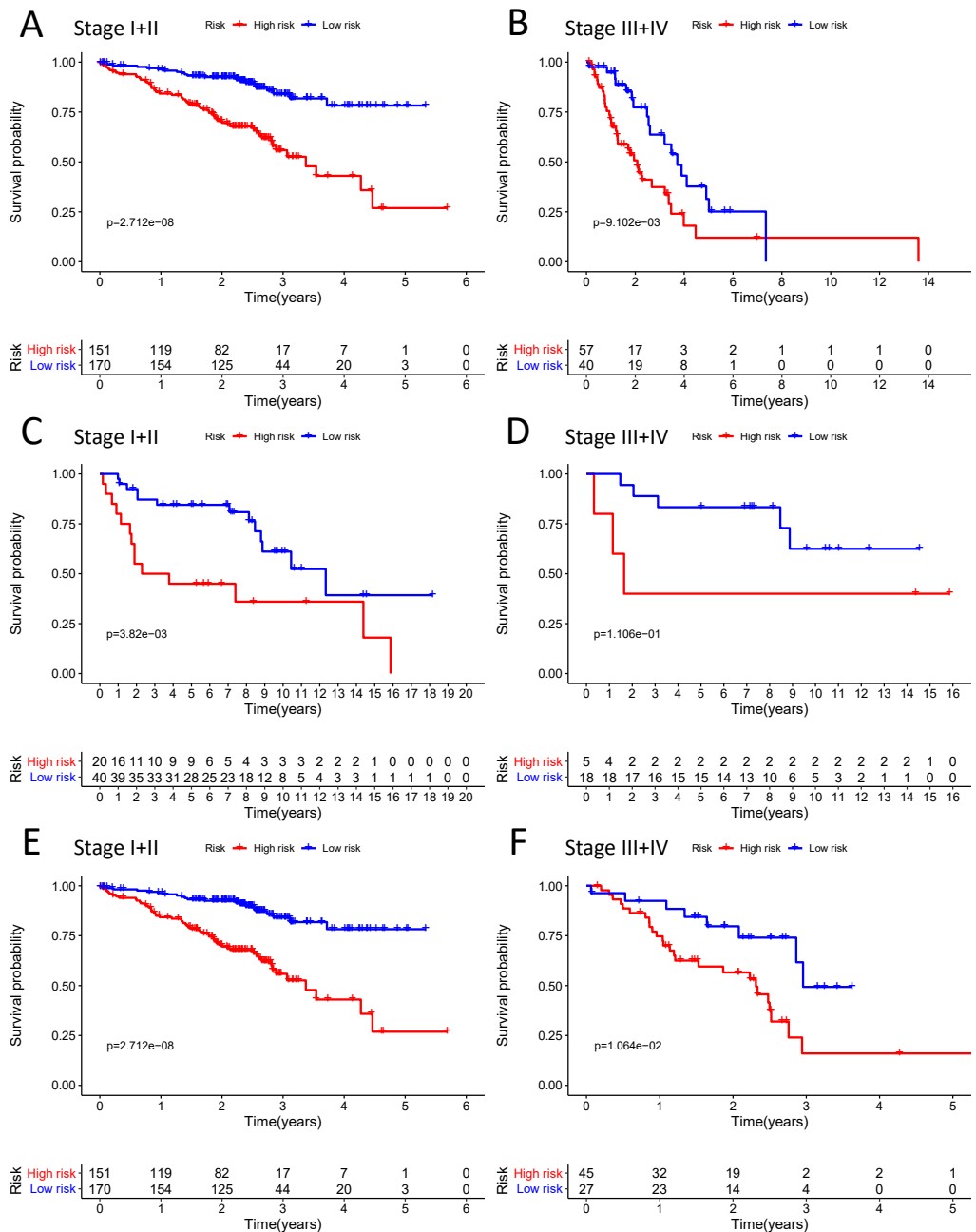

**Figure 8** **The Kaplan–Meier curve for the twenty-three gene signature in LUAD.** (A–B) The Kaplan–Meier curve showed that the survival of patients was significantly poorer in the high-risk group in TNM stages I and II/TNM stages III and IV of the TCGA-LUAD cohort; (C–D) the Kaplan–Meier curve showed that the survival of patients was significantly poorer in the high-risk group in TNM stages I and II/TNM stages III and IV of the GSE30219 cohort. However, there was no statistical significance in TNM stages III and IV; (E–F) the Kaplan–Meier curve showed that the survival of patients was significantly poorer in the high-risk group in TNM stages I and II/TNM stages III and IV of the GSE72094 cohort.

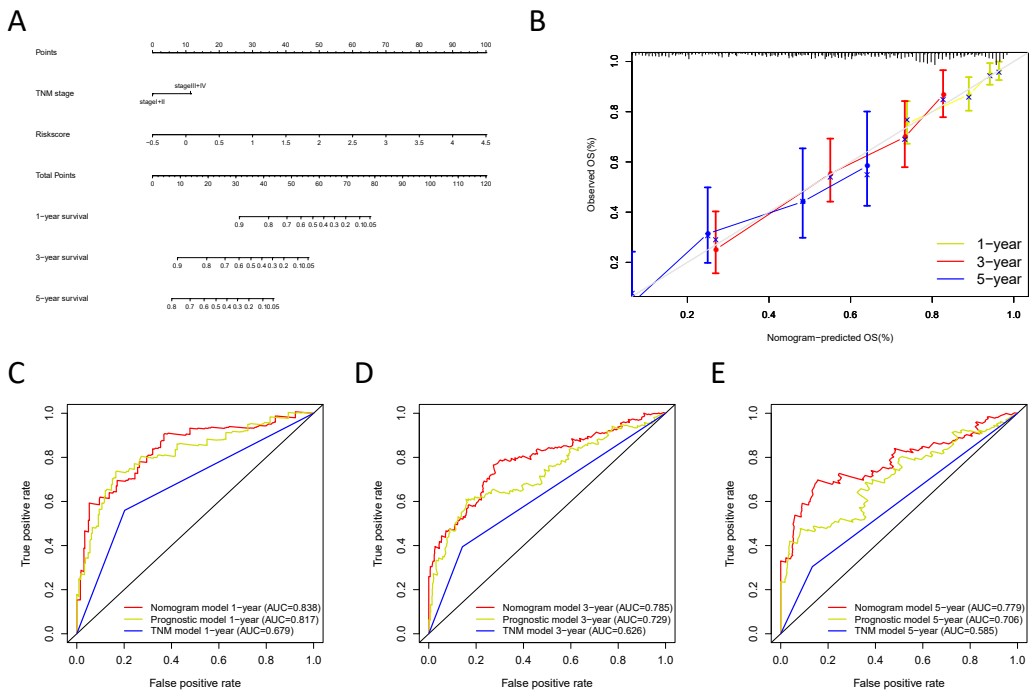

**Figure 9 Building the nomogram for predicting the overall survival of LUAD patients from the TCGA cohort.** (A) The nomogram plot was built based on all prognostic factors in the TCGA-LUAD; (B) the calibration plot for internal validation of the nomogram; (C–E) the time-dependent ROC curves of the nomograms compared for 1-, 3-, and 5-year overall survival in TCGA-LUAD.

ROC analysis from the TCGA-LUAD cohort and GEO cohorts verified the performance of the prognostic signature. Besides, the nomogram model showed better performance in prognosis predicting. We think that the nomogram will help us to make the clinical treatment strategy in the future. In conclusion, these results demonstrated a significant prognostic value of our prognostic model.

To better understand the molecular mechanism of metabolic genes, we enriched ten KEGG signalling pathways that are significantly related to metabolic gene models through GSEA. We found that patients in low-risk groups may profit from metabolic therapies. whereas, the outcomes provide feasible guidance for explaining the unknown mechanisms of labelling. GO analysis also indicated the twenty-three genes that were enriched in metabolic pathways. In conclusion, our signatures may reflect the disorder of LUAD patient tumour microenvironment and provide molecule biomarkers for treatment and prediction of the prognosis of LUAD.

In our twenty-three gene prognosis model, several metabolic genes, including GSTA3, ENTPD2, HK3, CHPT1, CTPS2, and ADCY9, were confirmed for the first time to be correlated with the prognosis of LUAD. GSTA3 is recognized as an antioxidative protease (*Chen et al., 2019a*). Studies of genetic analysis models and the mechanism of GSTA3 found that it was associated with tumour prognosis (*Bruzzoni-Giovanelli et al., 2015*). For example, GSTA3 was identified and regarded as a prognostic biomarker for nasopharyngeal

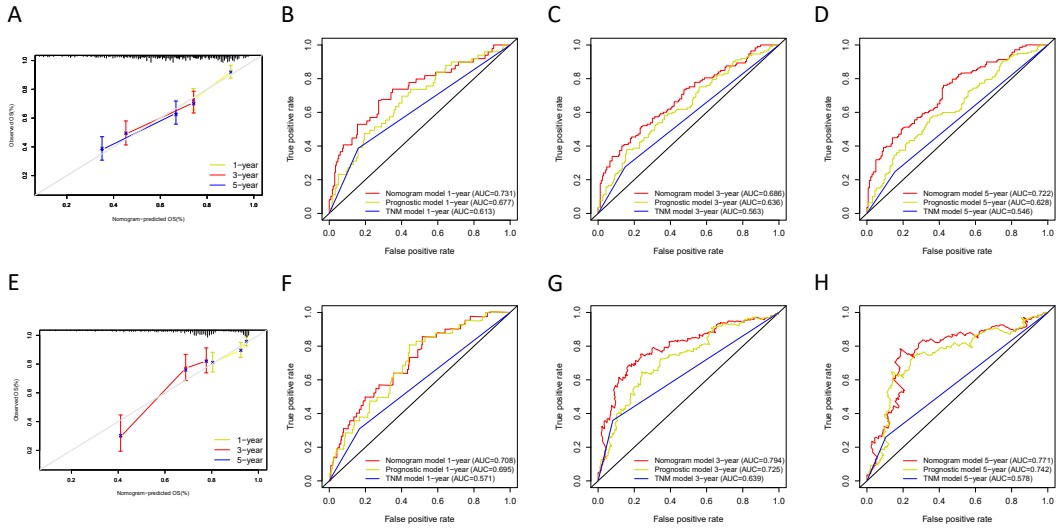

**Figure 10   Validation of the nomogram in the GSE30219 and GSE72094 cohorts.** (A) The calibration plot for validation of the nomogram in the GSE30219 cohort; (B–D) the time-dependent ROC curves of the nomograms compared for 1-, 3-, and 5-year overall survival in the GSE30219 cohort; (E) the calibration plot for validation of the nomogram in the GSE72094 cohort; (F–H) the time-dependent ROC curves of the nomograms compared for 1-, 3-, and 5-year overall survival in GSE30219 cohort.

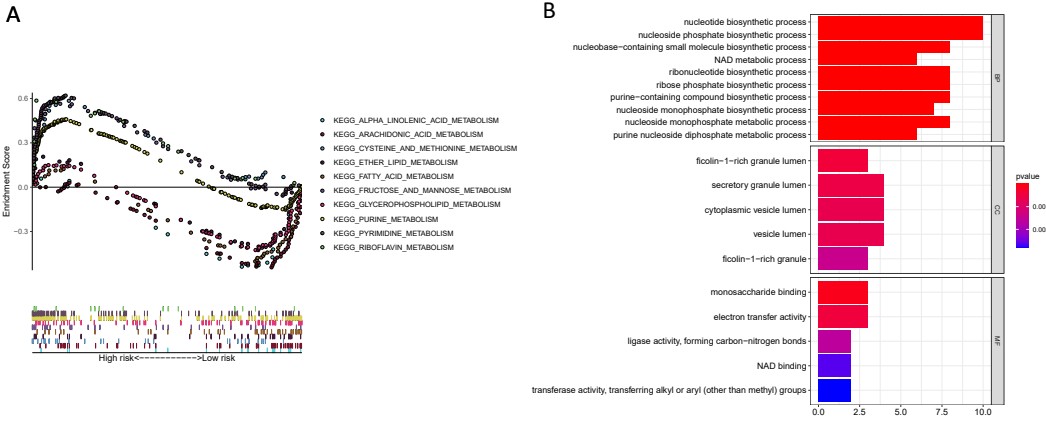

**Figure 11   KEGG and GO enrichment pathway analysis.** (A) KEGG enrichment pathway analysis of twenty-three metabolic-related genes obtained by the lasso Cox model: five representative KEGG pathways in high-risk patients. five representative KEGG pathways in low-risk patients. (B) GO enrichment pathway analysis of twenty-three metabolic-related genes obtained by the lasso cox model.

carcinoma and gastric cancer (*Duan et al., 2018*; *Zhang, Wu & Cheng, 2019*). GSTA3 inhibits HSC activation and liver fibrosis by inhibiting MAPK and GSK-3 $\beta$ signalling pathways, suggesting that GSTA3 could be a feasible target for therapeutic interventions for liver fibrosis (*Chen et al., 2019a*). GSTA3 overexpression in breast cancer cells stimulates proliferation and inhibits apoptosis, which leads to chemotherapy resistance and radiation resistance in tumour cells (*Thewes et al., 2010*). Overexpression of ENTPD2 can be regarded

**Table 6  The SIZE, ES, NES. NOM *p*-val, and FDR *q*-val of the KEGG enrichment pathways.**

| | SIZE | ES | NES | NOM P | FDR q-valve |
|---|---|---|---|---|---|
| Low risk group enrich pathways | | | | | |
| KEGG_ASTHMA | 28 | −0.802 | −2.039 | <0.01 | 0.040 |
| KEGG_ARACHIDONIC_ACID_METABOLISM | 58 | −0.558 | −1.956 | 0.002 | 0.057 |
| KEGG_GLYCEROPHOSPHOLIPID_METABOLISM | 77 | −0.435 | −1.739 | 0.008 | 0.117 |
| KEGG_ALDOSTERONE_REGULATED_SODIUM_ REABSORPTION | 42 | −0.494 | −1.702 | 0.010 | 0.135 |
| KEGG_ALPHA_LINOLENIC_ACID_METABOLISM | 19 | −0.568 | −1.672 | 0.019 | 0.136 |
| KEGG_FATTY_ACID_METABOLISM | 42 | −0.517 | −1.635 | 0.048 | 0.138 |
| High risk group enrich pathways | | | | | |
| KEGG_CELL_CYCLE | 124 | 0.746 | 2.384 | <0.01 | 0.001 |
| KEGG_PYRIMIDINE_METABOLISM | 98 | 0.620 | 2.160 | <0.01 | 0.008 |
| KEGG_MISMATCH_REPAIR | 23 | 0.852 | 2.149 | <0.01 | 0.005 |
| KEGG_UBIQUITIN_MEDIATED_PROTEOLYSIS | 134 | 0.541 | 2.144 | <0.01 | 0.004 |
| KEGG_P53_SIGNALING_PATHWAY | 68 | 0.546 | 2.100 | 0.002 | 0.006 |

**Notes.**
ES, enrichment score; NES, normalized enrichment score; NOM P, nominal *p* value; FDR q-valve, false discovery rate.

as a poor prognostic indicator of liver cancer. In an anoxic environment, ENTPD2 converts extracellular ATP into 5′-AMP, which inhibits the differentiation of myeloid inhibitory cells (MDSC) and promotes the stability of MDSC. The inhibition of ENTPD2 expression can reduce cancer growth and improve the effectiveness of immune checkpoint inhibitors (*Chiu et al., 2017*). HK3 is a very well-known glycolysis gene whose overexpression could be linked to hypoxia-induced upregulation of glycolysis and improvement in breast cancer cell survival (*Jarrar et al., 2020*). The pre-expression of HK3 is related to the epithelial-mesenchymal transition in colorectal cancer (CRC) and maybe a strategic metabolic gene for rapid proliferation, survival, and metastasis of CRC cells (*Pudova et al., 2018*). Besides, HK3 is associated with a CpG island methylated phenotype (CIMP) in colon adenocarcinoma (COAD). The upregulation of HK3 was reported in CIMP-high tumours compared to non-CIMP ones. HK3 can serve as a biomarker of high CIMP status in COAD (*Fedorova et al., 2019*). CHPT1 is considered to be a direct oestrogen receptor $\alpha$-regulatory gene and is necessary for oestrogen-induced choline metabolism. CHPT1 mediates metabolic changes in breast cancer cells, and silencing CHPT1 can inhibit breast cancer cell proliferation and early metastasis of tamoxifen-resistant breast cancer cells, suggesting that CHPT1 is a treatment target for cancers (*Fedorova et al., 2019*; *Jia et al., 2016*). CTPS2 had a profound effect on osteosarcoma metastasis (*Fan et al., 2019*) and also participated in the primary immunodeficiency of herpes virus susceptible populations (*Verzegnassi et al., 2018*). Low CTPS2 expression may be the underlying determinant of 5FU resistance (*Tan et al., 2011*). ADCY9 regulates signalling pathways mainly by producing a second messenger cyclic adenosine monophosphate. Some research found that ADCY9 acts as a key gene in the cisplatin response regulatory network in the pro-apoptotic stage in breast cancer treatment (*Fallahi & Godini, 2019*), and overexpression of ADCY9 is a poor

prognostic marker for disease-free survival in colon cancer (*Yi et al., 2018*). The above studies provided us with directions for studying these six genes in LUAD.

In addition to the above genes, many metabolic-related genes that were linked to the prognosis of the LUAD were included in seventeen additional genes, such as ALDH2, PKM, LDHA, and SMS. First, some of them, such as INPP4B, ALDOA, ALDH2, and LDHA, are significantly involved in LUAD suppression. INPP4B is regarded as a tumour suppressor of LC because it regulates the level of 3-phosphorylated phosphoinositide at the cellular level and activates phosphoinositide in PTEN-deficient cells (*Vo & Fruman, 2015*). ALDOA is a critical enzyme involved in metabolic reprogramming and metastasis of NSCLC that increases the ability of LC cells to migrate and invade by interacting with $\gamma$-actin. Blocking this interaction could be an effective cancer treatment (*Chang et al., 2019*). The expression level of ALDH2 was substantially correlated with a poor prognosis in LUAD (*Chen et al., 2018*). The principal role of ALDH2 is detoxifying acetaldehyde (ACE) to non-toxic acetic acid. ALDH2 inhibition leads to the accumulation of ACE, which enhances the migration of LUAD cells by damaging DNA. Therefore, activating ALDH2 could provide a novel strategy for treating LUAD (*Li et al., 2019*). LDHA is an essential enzyme for glucose metabolism. It can inhibit the expression of HIF-1$\alpha$ and its downstream gene GLUT1 and thus inhibit the growth of NSCLC cells (*Massari et al., 2016*). Some additional genes are associate with tumour treatment tolerance. For example, low expression of MOBA inhibits the NF-$\kappa$B signalling pathway, leading to NSCLC radioresistance (*Son et al., 2016*). In another study, silencing TYMS increased the sensitivity of NSCLC tumour cells to pemetrexed (*Agulló-Ortuño et al., 2020*). In addition, several metabolic genes have also been confirmed to be closely related to the occurrence of LC and have provided some novel research directions. For instance, B4GALT1 is related to aberrant gene promoter methylation and maintains the stemness of LC stem cells (*Zhang, Zhang & Yu, 2019*). SMS participates in the lymphatic metastasis of LUAD (*Lemay et al., 2019*).

Compared to other studies, we first used metabolic-related genes to build a prognosis model from TCGA-LUAD and validated it in two GEO datasets. Our gene signature also had a better prediction ability compared to the other model. This risk model perhaps provides potential biomarkers for studying the relationship of metabolic microenvironmental diseases, metabolic therapies, and therapeutic responses. The risk model and TNM model were used to build the nomogram, and the calibration plot, C-index, ROC analysis validated the clinical application of the twenty-three metabolic-related gene signature, which may be helpful for the diagnosis, treatment, and prognosis of LUAD.

However, there are still some limitations in our study: first, the connection between 23 metabolic gene markers and the metabolic microenvironment needs to be verified by basic experiments. Secondly, a large number of clinical samples are lacking to verify whether the prognostic effect of metabolic therapy is related to its metabolic microenvironment. Basic and clinical trials will need to continue in the future to explore this relationship. Besides, some prognostic metabolic genes may not meet the screening criteria and were not included when constructing the prognostic signature, which could also lead to the development and progression of LUAD. Based on the above factors, the application of risk score to the clinic remains a huge challenge

## CONCLUSION

In conclusion, our research identified a 23 metabolic-related gene signature for LUAD patient prognosis based on the TCGA data set. Our signature provides potential biomarkers for studying aspects of metabolic microenvironmental diseases, metabolic therapies, and therapeutic responses. However, it is still urgent to further investigate the relationship between metabolic microenvironment and metabolic therapy, and more functional experiments are required for revealing the mechanism of metabolic genes in the process of LUAD development.

## ACKNOWLEDGEMENTS

The authors thank Professor Yongguang Tao for his comments and suggestions throughout the writing process. It is with regret that not all relevant studies could be cited due to space limitations.

### Funding

This work was supported by the Hunan Provincial Key Area R&D Programmes (2019SK2253) and the National Natural Science Foundation of China (81672308, X. Wang; 81672787, Y. Tao). The funders had no role in study design, data collection and analysis, decision to publish, or preparation of the manuscript.

### Grant Disclosures

The following grant information was disclosed by the authors:
The Hunan Provincial Key Area R&D Programmes: 2019SK2253.
National Natural Science Foundation of China: 81672308, 81672787.

### Competing Interests

The authors declare there are no competing interests.

### Author Contributions

- Zhenyu Zhao conceived and designed the experiments, performed the experiments, analyzed the data, prepared figures and/or tables, authored or reviewed drafts of the paper, and approved the final draft.
- Boxue He, Qidong Cai, Pengfei Zhang, Xiong Peng, Yuqian Zhang, Hui Xie and Xiang Wang conceived and designed the experiments, authored or reviewed drafts of the paper, and approved the final draft.

### Data Availability

The raw measurements are available in the Supplemental Files. The training cohort data is available in at TCGA (https://portal.gdc.cancer.gov/projects/TCGA-LUAD"). The testing cohort data is available at NCBI GEO: GSE30219 and GSE72094.

## Supplemental Information

Supplemental information for this article can be found online at http://dx.doi.org/10.7717/peerj.10008#supplemental-information.

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
