# Peer review of "A model of twenty-three metabolic-related genes predicting overall survival for lung adenocarcinoma"

_PeerJ, doi:10.7717/peerj.10008_

## Round 0.1 · original submission · Major Revisions

When revising your manuscript, please consider all issues mentioned in the comments from the three reviewers carefully and provide suitable responses for any comments. Please note that your revised submission may need to be re-reviewed.

PeeJ values your contribution and I look forward to receiving your revised manuscript.

Reviewer 1 ·

Basic reporting

no comment

Experimental design

no comment

Validity of the findings

no comment

Additional comments

The Zhao et al. contrasted a model of twenty-four metabolic-related genes predicting overall survival for lung adenocarcinoma, but some questions were needed to be optimized:
1. More detailed description (such as the packages and functions) should be given for readers for the three-step method in the Materials and Methods section.
2. The order of subtitles should be corrected, such as 2.6, 2.8 …
3. Authors should provide clearer figures for readers.
4. Authors use LUAD RNA-Seq data and clinical data from The Cancer Genome Atlas (TCGA) were regarded as a training cohort, and microarray data from GEO (GSE30219) was regarded as a testing cohort and utilized for validation. Authors should interpret the advantages and limitations of this experimental design.
5. Authors should perform the Gene Ontology analysis and interpret the biological function.
6. The grammar and English style were needed to be optimized.
7. More references are needed in the Introduction section.

Reviewer 2 ·

Basic reporting

no comment

Experimental design

no comment

Validity of the findings

no comment

Additional comments

The authors identified 24 metabolic-related genes predicting overall survival for lung adenocarcinoma (LUAD) by existing statistic methods in this paper. The method of this paper is not a new method, and the results are not exciting.
The authors identified the 24 metabolic-related genes by TCGA dataset with 445 samples, and verified the results by a GEO dataset with 83 sample? I think that the authors should add more independent datasets to verified the validation of the 24 metabolic-related genes.
The method section in this paper is not clear, and I suggest that the authors should re-write the method section and make the method thread clearly. A flowchart for calculation process can help reader to understand the method section of this paper.

The authors should explain some items in the paper, for example:
What’s the meaning of “logFC” in line 87?
The authors need explain the meaning of the “Coefficient” in line 92?
What kind of index is “C-index” in line 216?
What’s the meaning of “CoefficientmRNA1” and “CoefficientmRNA2”?

Reviewer 3 ·

Basic reporting

The language should be improved

Experimental design

Fair

Validity of the findings

Fair

Additional comments

The authors proposed 24 key genes of LUAD , and then built the model based on these 24 genes. This is an interesting paper. However, there are some problems.
1. the authors applied survival curve to test the proposed genes. This is not enough. As the sample is limitted, the validatoins should be performed from different aspects.
2.Figure 4 is unclear.
3.The authors should compared their model with referencec.

---

## Round 0.2 · Minor Revisions

I am pleased to inform you that your submission entitled “A model of twenty-three metabolic -related genes predicting overall survival for lung adenocarcinoma” has been provisionally accepted for publication in PeerJ.

However, before your paper can be formally accepted and forwarded to our Production Department, you are requested to make the corrections indicated below.

Please write the date to download the cohort date from TCGA and GEO database, the data to perform the bioinformatics analysis using the web tools such as KEGG and Go analysis. Please note the full name of HR, SD etc under Table 1, Table 2, and Table 3 and unified the correct form for P (P, p, p-value) and correct a few errors in the manuscript such as 0686 in line 243 on page 8, "c2. cp. kegg. v6.2. symbols. gmt. gene sets" VS “ggplot2, gridExtra, grid, plyr” VS ‘clusterProfiler, org.Hs.eg.db, plot, ggplot2’ (different quotation marks), the literature in line 534 on page 16 and so on. In section 2.7, a text formula should be better than the image formula there.

We look forward to receiving your final version of your manuscript.

Reviewer 1 ·

Basic reporting

no comment

Experimental design

no comment

Validity of the findings

no comment

Additional comments

The authors have addressed all of the comments/suggestions raised by the reviewers.

Reviewer 3 ·

Basic reporting

The MS has been well revised, and the authors' responds are satfified

Experimental design

OK

Validity of the findings

OK

Additional comments

The MS has been well revised, and the authors' responds are satfified

---

## Round 0.3 · accepted · Accept

A few mistakes still need to be corrected DURING THE PROOFREADING STAGE. For examples: line 170: "24 genes" to "23 genes"; line 172: deleted "INPP5A, "; line 244: "0686" to "0.686" which I mentioned for a second time. Please do things more carefully.